# Incidence, Outcomes and Risk Factors of Recurrent Ventilator Associated Pneumonia in COVID-19 Patients: A Retrospective Multicenter Study

**DOI:** 10.3390/jcm11237097

**Published:** 2022-11-30

**Authors:** Ines Gragueb-Chatti, Hervé Hyvernat, Marc Leone, Geoffray Agard, Noémie Peres, Christophe Guervilly, Mohamed Boucekine, Dany Hamidi, Laurent Papazian, Jean Dellamonica, Alexandre Lopez, Sami Hraiech

**Affiliations:** 1Service de Médecine Intensive-Réanimation, AP-HM, Hôpital Nord, 13015 Marseille, France; 2Service de Médecine Intensive-Réanimation, CHU Nice, 06202 Nice, France; 3Service d’Anesthésie et de Réanimation, Assistance Publique Hôpitaux de Marseille, Aix Marseille Université, 13015 Marseille, France; 4Service de Réanimation Polyvalente, Centre Hospitalier Intercommunal Toulon—La Seyne sur Mer, 83056 Toulon, France; 5Health Service Research and Quality of Life Center (CEReSS), Aix-Marseille Université, 27 Boulevard Jean-Moulin, 13005 Marseille, France

**Keywords:** COVID-19, ICU, ventilator-associated pneumonia, acute respiratory distress syndrome, recurrence of VAP

## Abstract

Background: High incidence of ventilator associated pneumonia (VAP) has been reported in critically ill patients with COVID-19. Among these patients, we aimed to assess the incidence, outcomes and risk factors of VAP recurrences. Methods: We conducted an observational retrospective study in three French intensive care units (ICUs). Patients admitted for a documented COVID-19 from March 2020 to May 2021 and requiring mechanical ventilation (MV) for ≥48 h were included. The study main outcome was the incidence of VAP recurrences. Secondary outcomes were the duration of MV, ICU and hospital length of stay and mortality according to VAP and recurrences. We also assessed the factors associated with VAP recurrences. Results: During the study period, 398 patients met the inclusion criteria. A total of 236 (59%) of them had at least one VAP episode during their ICU stay and 109 (46%) of these patients developed at least one recurrence. The incidence of VAP recurrence considering death and extubation as competing events was 29.6% (IC = [0.250–0.343]). Seventy-eight percent of recurrences were due to the same bacteria (relapses). Patients with a VAP recurrence had a longer duration of MV as compared with one VAP and no VAP patients (41 (25–56) vs. 16 (8–30) and 10 (5–18) days; *p* < 0.001) and a longer ICU length of stay (46 (29–66) vs. 22 (12–36) and 14 (9–25) days; *p* < 0.001). The 90-day mortality was higher in the recurrence group as compared with the no VAP group only (31.2 vs. 21.0% (*p* = 0.021)). In a multivariate analysis including bacterial co-infection at admission, the use of immunosuppressive therapies and the bacteria responsible for the first VAP episode, the duration of MV was the only factor independently associated with VAP recurrence. Conclusion: In COVID-19 associated respiratory failure, recurrences affected 46% of patients with a first episode of VAP. VAP recurrences were mainly relapses and were associated with a prolonged duration of MV and ICU length of stay but not with a higher mortality. MV duration was the only factor associated with recurrences.

## 1. Background

Patients admitted to the intensive care unit (ICU) because of severe forms of Coronavirus Disease 2019 (COVID-19) require invasive mechanical ventilation (MV) in up to 80% of cases [1]. Unexpectedly high rates of ventilator associated pneumonia (VAP) have been reported among these patients, reaching 50 to 80% according to the series [2,3]. Specific features of SARS-CoV-2 pneumonia seem to be involved as COVID-19 patients develop more VAP than do Influenza patients, regardless the duration of MV [3]. Several mechanisms have been suggested to explain this increase in nosocomial bacterial pneumonia, mainly the immune system impairment due to SARS-CoV-2 infection [4,5], the use of corticosteroids [5,6,7], and the high incidence of ARDS with prolonged MV and recourse to prone positioning [8,9]. 

Some series also pointed out the recurrence of several episodes of VAP in the same patients [10,11]. Recurrence was most often due to the same pathogen (relapse), despite a well conducted antibiotic treatment [10] and was associated with a poor prognosis. However, these data were limited to patients under veno-veinous extracorporeal membrane oxygenation (ECMO). No study specifically addressed the question of VAP recurrences in patients with severe SARS-CoV-2 pneumonia despite the use of broad spectrum and prolonged antibiotic treatments they are associated with [10,11,12].

We aimed to describe clinical and microbial characteristics of patients with a VAP recurrence during COVID-19. Primary outcome was to determine the incidence of VAP recurrences. Secondary outcomes were to describe their microbiological features, evaluate their impact on the duration of MV, ICU and hospital length of stay and mortality and to bring to light potential risk factors exposing to VAP recurrence. 

## 2. Methods

### 2.1. Study Design and Population 

We conducted an observational retrospective study in three ICUs from two university hospitals in Southern France. From 1 March 2020 to 1 May 2021, all patients aged 18 or older, admitted for acute respiratory failure related to a documented SARS-CoV-2 pneumonia (from a nasopharyngeal or pulmonary sample RT-PCR) and requiring invasive MV for at least 48 h were included. Patients for whom withholding of treatments was decided during the first 48 h after ICU admission, aged under 18, deprived of liberty or without social protection, refusing (patients or relatives) the use of medical data collected for routine care were not included. 

### 2.2. Definitions

#### VAP Diagnosis 

VAP was diagnosed in patients having received MV for at least 48 h when the following criteria were met [13,14]:-New or progressive persistent infiltration on chest radiograph;-At least two of the following: new onset of fever, purulent endotracheal aspirate, leukocytosis or leucopenia, increased minute ventilation, arterial oxygenation decline, need for increased vasopressor infusion to maintain blood pressure (for patients with ARDS, for whom demonstration of radiologic deterioration is difficult, at least two of the preceding criteria sufficed);-A positive quantitative or qualitative culture from broncho-alveolar lavage (BAL), protected distal sample (PDS) or endotracheal aspirate (ETA).

A positive bacterial culture on a respiratory sample without clinical sign of pneumonia and without antibiotic treatment initiated was considered as a colonization. 

### 2.3. Bacterial Co-Infection at ICU Admission

A bacterial co-infection was diagnosed when an invasive (BAL, PDS, ETA, blood cultures) or non-invasive (sputum sample, multiplex PCR, *Streptococcus pneumoniae* or *Legionnella pneumophila* antigenuria) sample was positive before ICU admission or within the 48 h following it.

### 2.4. Relapse and Recurrence of VAP

Recurrence of VAP was defined as a new onset following a regression of clinical signs (fever, expectorations, and vasopressor infusion), inflammatory biomarkers and infiltration on chest radiography after a complete adequate antibiotic treatment (at least one antibiotic active against the documented bacteria). VAP recurrence was diagnosed on clinical signs reappearance and at least one bacterial species growth at a significant concentration from respiratory samples. Relapse was defined as a recurrence involving at least one of the initial causative bacteria; otherwise, it was considered a superinfection.

### 2.5. Baseline Assessment and Data Collection

Data were collected from the patients’ electronic medical file. Demographic characteristics, comorbidities, severity at ICU admission, date of SARS-CoV-2 RT-PCR positivity, date of ICU admission, date of intubation and invasive MV, need for ECMO, antiviral treatment, initial bacterial co-infection and antibiotics received at ICU admission, VAP with microbiological documentation and recurrences, antibiotics received during the ICU stay, duration of antibiotics treatment, duration of total invasive MV, ICU and hospital stay, status at day 28 (from the start of ICU hospitalization), day 90 (includes after hospital discharge), ICU and hospital mortality were obtained. The use of immunomodulatory/immunosuppressive (IS) therapies was also recorded: Dexamethasone (at 6 or 12 mg per day) [15]Methylprednisolone for persistent ARDS as described elsewhere [16]Hydrocortisone (at 200 mg per day)Interleukine-6 (Il-6) receptor antagonist (Tocilizumab) [17]Interleukin-1(II-1) receptor antagonist (Anakinra) [18]Janus Kinases (JAK) receptor antagonist (Ruxolitinib) [18]The combination of several of them during the same ICU stay

### 2.6. Antibiotic Treatment 

Empiric antibiotic therapy was started in case of VAP suspicion according to national and international recommendations [19,20,21,22]. De-escalation was performed if possible as soon as the results of microbiological investigations performed were available [21,23]. 

### 2.7. Management of Antibiotic Treatment 

Antibiotic administration through prolonged infusions was used as it was part of the routine care in each of the three ICUs. Empirical antibiotic treatment was considered adequate when the patient received at least one antibiotic active against the responsible pathogen [23].

Therapeutic drug monitoring was performed according to physicians’ decision.

### 2.8. Study Outcomes

The primary outcome was the incidence of VAP recurrences. The secondary outcomes were the microbiological description of VAP recurrences, the percentage of antibiotic target attainment (serum concentration above 4 times the minimal inhibitory concentration (MIC) of documented bacteria), the evolution towards lung abscess, the impact of VAP and recurrences on the duration of invasive MV, ICU and hospital length of stay and mortality and the factors associated with VAP recurrences. 

### 2.9. Statistical Analysis 

Statistical analysis was performed using SPSS Version 20 (IBM SPSS Inc., Chicago, IL, USA). 

Continuous variables are expressed as mean ± standard deviation (SD) or as median with interquartile range, and categorical variables are reported as count and percentages. Comparisons between groups were performed using Student’s *t*-test or Mann–Whitney U as appropriate. Comparisons of percentages were performed using Chi-square test or (Fisher’s exact test, as appropriate). 

We performed a multivariate logistic regression analysis including non collinear variables with *p* < 0.2 in univariate analysis to determine the influence of clinical parameters on VAP recurrence. 

Fine–Gray model was used to estimate the cumulative incidence of VAP recurrence considering death and extubation as competing risks [24]. Analysis was performed using the cuminc function from cmprsk r package.

We confirmed impact of variables on timing of VAP incidence by a COX model and constructed Kaplan–Meier curves. Curves were compared with the Log Rank test. 

The statistical significance was defined as *p* < 0.05.

## 3. Results

### 3.1. Patients’ Characteristics at ICU Admission 

Study flow chart is presented in Figure 1. A total of 398 patients were included in the final analysis. Table 1 shows the repartition of patients according to the occurrence of VAP. During the ICU stay, 162 (40.7%) patients did not develop VAP (no VAP group), 127 (31.9%) had a single VAP episode (1 VAP group) and 109 (27.4%) had a recurrence of VAP (2 or more episodes, recurrence group). A total of 236 (59%) patients had at least one VAP and 109 (46%) of these patients developed at least one recurrence (65 patients had 2 VAP and 44 patients had 3 VAP). The recurrence was diagnosed using BAL and ETA in 44 and 65 cases, respectively. In the recurrence group, the median delay from first to second VAP was 11.7 [5.0–17.0] days. Admissions were spread during the first three waves of pandemics in France. A total of 264 (66.3%) patients received empirical antibiotics at ICU admission without any difference between groups. An initial co-infection was documented in only 44 (11.1%) patients and was more frequent in patients that presented a VAP recurrence than in one VAP or no VAP groups (17.4% vs. 10.2% and 7.4% respectively; *p* = 0.035). In the recurrence group, V/V ECMO was more frequently used than in the one VAP and no VAP patients (34.9% vs. 14.2% and 13.6%; *p* < 0.001).

ECMO: extracorporeal membrane oxygenation; ICU: intensive care unit; IQR: interquartile range; IMV: invasive mechanical ventilation; SAPS II: simplified acute physiologic Score II; SD: standard deviation; SOFA: sequential organ failure assessment; VAP: ventilator-associated pneumonia.

### 3.2. VAP and Recurrence Incidence

Figure 2 shows the cumulative incidence of VAP recurrence, considering death and duration of MV (extubation) as competing events. The incidence of VAP recurrence was 29.6% (IC = [0.250–0.343]).

### 3.3. Use of Immunosuppressive Therapies during the ICU Stays

Table 2 shows the use of IS therapies in each group. Patients in the recurrence group were more often treated with methylprednisolone for persistent ARDS and received more frequently a combination of two IS as compared with 1 VAP group and no VAP group (*p* < 0.01). 

### 3.4. Microbiological and Pharmacological Results 

Table 3 depicts the micro-organisms responsible of VAP. Gram-negative bacteria (55.9%), especially *Enterobacteriaceae*, were predominant during the first VAP episode. Non-fermenting Gram-negative bacteria (*Pseudomonas aeruginosa* and *Stenotrophomonas maltophilia*) were majority during recurrences (54.2% of gram-negative bacilli). Gram-positive pathogens (25.7%) were mainly methicillin-susceptible *Staphylococcus aureus* (MSSA) and *Enterococcus* spp. 

Seventy-eight percent of recurrences were relapses—i.e., involved the same bacteria—despite appropriate treatment of the preceding VAP. 

Therapeutic drug monitoring was performed in 69 (54%) of 127 patients during first VAP episode. Serum antibiotic concentrations reached therapeutic range according to MIC90 in 50 (72.5%) patients. Noteworthy, the patients with VAP recurrence developed more frequently lung abscesses, as compared with those with one VAP episode (16 (14.7%) vs. 8 (6.2%); *p* < 0.001).

Multi-drug resistant is defined as non-susceptibility to ≥1 drug in ≥3 antimicrobial categories.

### 3.5. Clinical Outcomes 

Table 4 shows the clinical outcomes of patients according to the occurrence of VAP and recurrence. A total of 19 patients were lost to follow-up. The recurrence group had increased duration of invasive MV (41 (25–56) vs. 16 (8–30) and 10 (5–18) days; *p* < 0.001) and ICU stay duration as compared with one VAP and no VAP groups (46 (29–66) vs. 22 (12–36) and 14 (9–25) days; *p* < 0.001). The 90-day mortality was higher in the recurrence group as compared with the no VAP group, 31.2 vs. 21.0% (*p* = 0.021). There was no mortality difference between the recurrence group and the 1 VAP group (*p* = 0.41). 

The predicted probability of death at day 90 was 33.9% (IC [26%; 44.2%]) for a patient with no VAP, 33.5% (IC [25.6%; 43.9%]) with one VAP episode and 47.8% (IC [21.9%; 100%]) for a patient with a VAP recurrence.

### 3.6. Factors Associated with VAP Recurrences

Factors associated with VAP recurrence were evaluated among patients with at least one VAP episode (one VAP and recurrence groups, *n* = 236). Age, SAPS2 score, SOFA score at ICU admission, obesity, bacterial co-infection at ICU admission, IS therapy, antibiotic target attainment, type of bacteria responsible for the first VAP, and duration of invasive MV prior the first VAP were included in the univariate analysis. Variables that reached *p* values of less than 0.20 in univariate analysis were included in the multivariate analysis (Table 5).

The duration of MV was the only variable independently associated with VAP recurrence. The specific role of IS therapies on the timing of second VAP occurrence was assessed using a Cox regression model. We used univariate Cox model testing: (a) all IS therapies, (b) only steroidal IS or (c) only non-steroidal IS on delay of VAP relapse. The use of steroidal IS (i.e., dexamethasone or hydrocortisone or methylprednisolone) delayed the second VAP by a mean of 5 days (20.0 [17.7–22.2] vs. 14.7 [12.6–16.7] days; *p* = 0.002) (Figure 3a). Non-steroidal IS treatment shortened the delay of second VAP occurrence by a mean of 5 days (15.1 [12.6–17.7] vs. 20.0 [17.8–22.2] days; *p* = 0.006) (Figure 3b).

Concerning the produced product of the time (delay of VAP relapse) by the covariable, *p* value was, respectively, 0.276 for all IS, 0.923 for steroidal IS and 0.220 for non-steroidal IS, indicating for all models no gross violation of proportional hazard of Cox model.

## 4. Discussion

In this cohort specifically addressing the question of VAP recurrences during COVID-19 pneumonia, more than half of the patients developed at least one VAP episode and 46% of these patients had at least one recurrence. The incidence of VAP recurrence considering death and extubation as competing events was 29.6%. *Enterobacteriaceae* and non-fermenting Gram-negative bacteria were mainly involved and 78% of VAP recurrences were relapses. Recurrences were associated with longer duration of MV and ICU length of stay, although 90-day mortality was not affected. The duration of MV was the only factor independently associated with recurrences, even after considering the use of immunosuppressive therapies.

The high rate of VAP described in our series is in line with a recent review showing that in COVID-19 patients, VAP incidence ranged from 21 to 85% [12]. In a large European cohort, Rouzé et al. [3] reported a 51% incidence, significantly higher than in Influenza patients. As a comparison, a 29% rate of VAP in non-COVID-19 ARDS patients was described [21]. Few data are available on VAP recurrences, with rates ranging from 8 to 25% [3,25,26,27,28] in studies not designed to explore specifically this endpoint. In a highly selected population of patients under V/V ECMO, Luyt et al. reported up to 59% of recurrences [10].

We found that prolonged invasive MV was the only factor independently associated with the risk of VAP recurrence. Although it is difficult to characterize the causal relationship between VAP and MV duration, several studies showed that COVID-19 patients have an increased risk of VAP, independently of the duration of MV [3,10,28]. We assessed here the role of immunosuppressive treatments. Previous studies suggested that dexamethasone alone was not associated with an increased risk of VAP [2]. In our cohort, the association of two IS therapies was used in 48.5% of patients, mainly a combination of dexamethasone and tocilizumab and/or methylprednisolone for persistent ARDS. However, neither the treatment with one nor the combination of several IS were independently associated with an increased VAP recurrence risk. This is of particular interest considering that dexamethasone was a part of standard of care and tocilizumab was largely used in ICU patients [6,15,17,29]. Noteworthy, the use of non-steroidal IS therapies was associated with an earlier development of VAP recurrence. IL-6 antagonists cause a transient but long-lasting immunosuppressive state, which may favor the occurrence of bacterial superinfections, such as VAP. Conversely, the use of steroids was associated with a delayed recurrence of VAP.

As described in the series from Luyt et al. [10], we found that 78% of recurrences were relapses, mainly involving *Enterobacteriaceae*. This result questions the efficacy of first VAP antibiotic treatment. However, when therapeutic drug monitoring was performed, target attainment was reached in 72.5% of patients. It has been suggested that pulmonary vascular endothelial inflammation and subsequent thrombosis might make the lung parenchyma a favorable substrate for bacterial growth and prevent antimicrobial penetration [30,31]. Altogether, our findings highlight the need for secondary infection monitoring, [25,26,27]. The high rate of relapses in our patients also questions about the best duration of antibiotic treatment in COVID-19 patients with bacterial co-infection. This seems a critical issue since we observed an unexpected high number of lung abscesses (14.7%) in the recurrence group, also reported in a previous cohort [32]. In our series, all patients with a first VAP were treated for 7 consecutive days, as recommended [33,34,35,36]. The so-called COVID-19 related “immunoparalysis” [37] could also explain the high rate of relapses. Decreased mHLA-DR expression is associated with the development of severe respiratory failure, and presumably may contribute to pronounced susceptibility to bacterial superinfections [34].

VAP recurrence was associated with a prolonged invasive MV duration and ICU length of stay although it did not affect 90-day mortality. Previous studies have shown that VAP during COVID-19 ARDS were associated with a higher mortality [38,39]. As it has been proposed in non-COVID-19 patients, VAP seems associated with prolonged duration of invasive MV and prolonged ICU stay, whereas mortality is mainly driven by patients’ underlying conditions and illness severity [8].

Our study has several limitations. First, the retrospective design with inherently associated bias. Second, the low rate of patients with serum antibiotic concentration monitoring prevents determining the effect of under-dosing in relapses. Finally, the strong association between patient’s severity, duration of MV and the use of immunosuppressive treatments hardens to strongly conclude about VAP recurrence risk factors. In our analysis, the weight of invasive MV duration over-rode other variables. In particular, the role of immunosuppressive therapies deserves to be more deeply explored.

## 5. Conclusions

In this series, we found that nearly half of patients under invasive MV for COVID-19 pneumonia with a first VAP episode developed recurrences, which were relapses in most cases. Patients with a VAP recurrence had a longer duration of invasive MV and ICU length of stay but not a higher mortality. MV duration was the only factor associated with VAP recurrences.

## Figures and Tables

**Figure 1 jcm-11-07097-f001:**
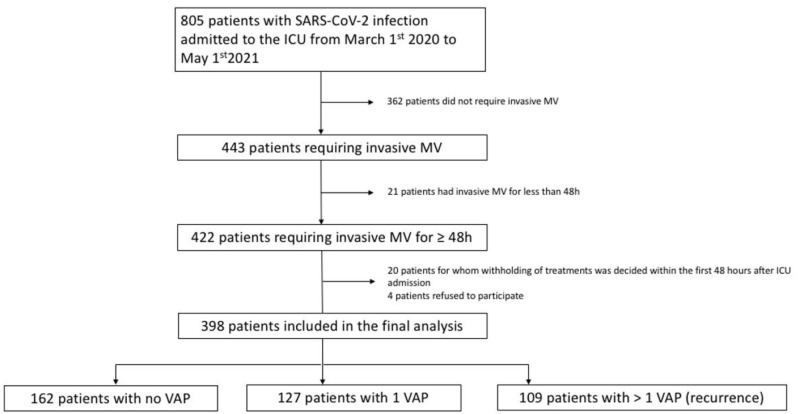
Study flow chart. ICU: intensive care unit; MV: mechanical ventilation; VAP: ventilator-associated pneumonia.

**Figure 2 jcm-11-07097-f002:**
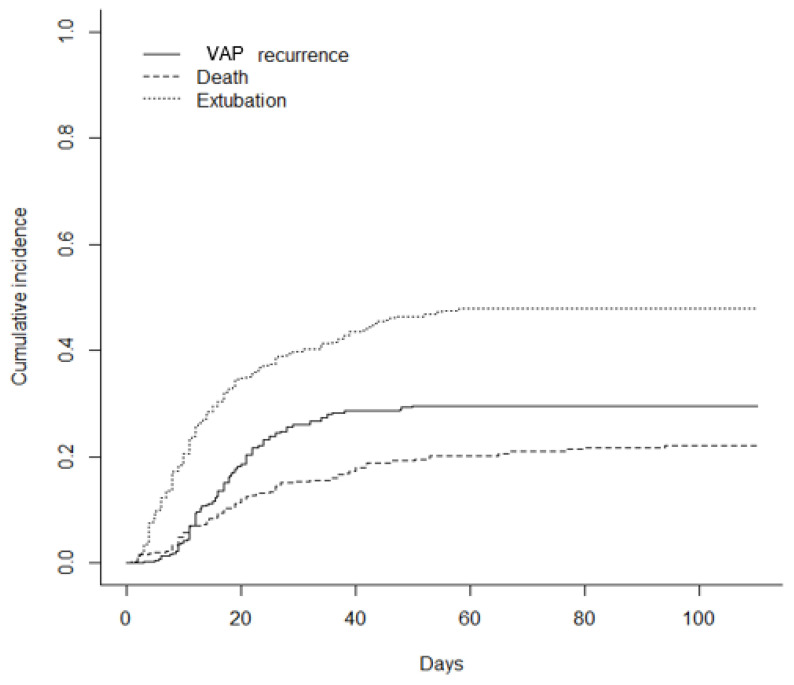
Estimated cumulative incidence of ventilator-associated pneumonia (VAP) recurrence considering death and extubation as competing events. VAP: ventilator-associated pneumonia.

**Figure 3 jcm-11-07097-f003:**
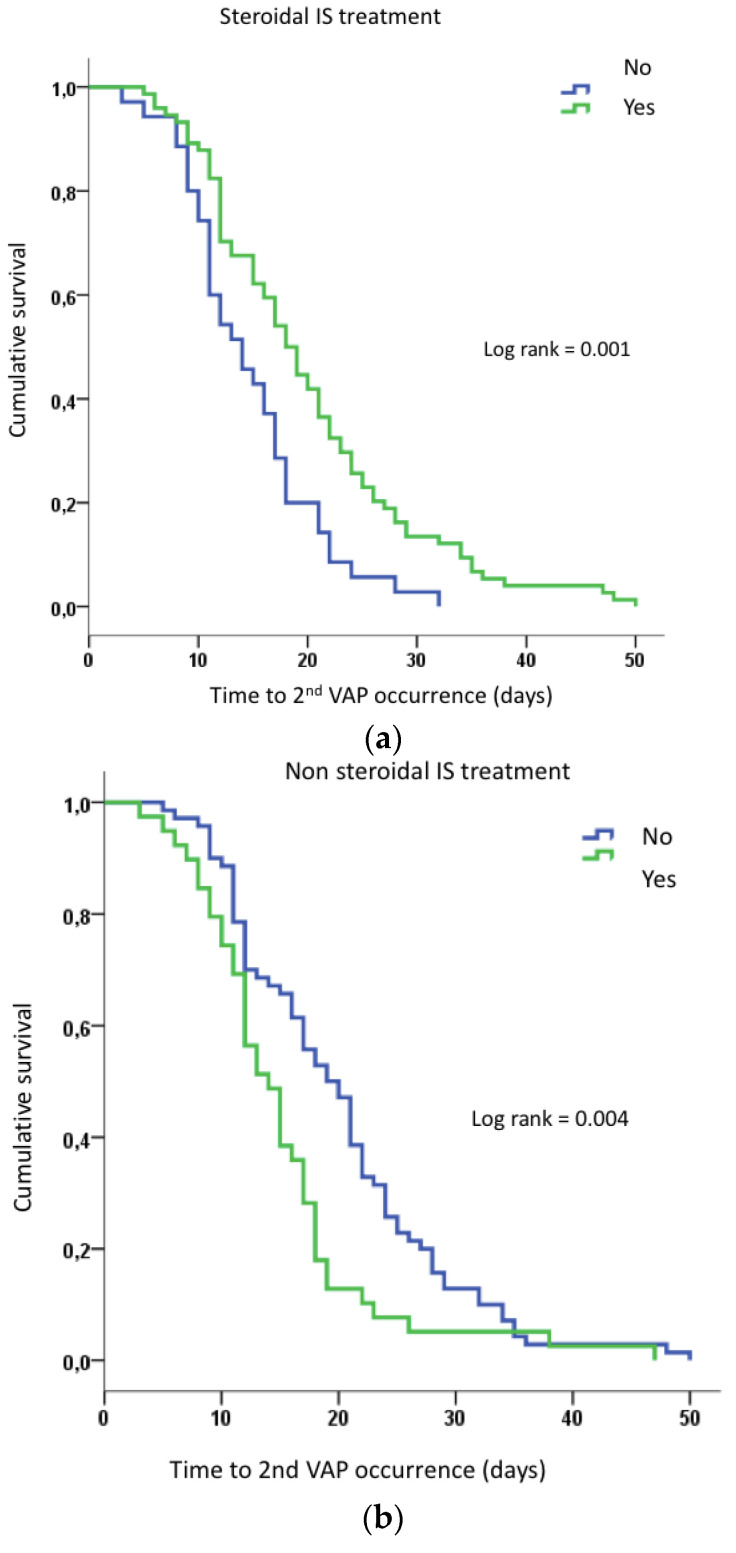
(**a**): Cumulated survival probability before 2nd VAP occurrence according to the use of steroidal IS treatment (subjects were censored at the time of VAP recurrence). (**b**): Cumulated survival probability before 2nd VAP occurrence according to the use of non-steroidal IS treatment (subjects were censored at the time of VAP recurrence). IS: immunosuppressive; VAP: ventilator-associated pneumonia.

**Table 1 jcm-11-07097-t001:** Patients’ main characteristics at ICU admission according to the occurrence of VAP.

	TOTAL (*n* = 398)	0 VAP (*n* = 162)	1 VAP (*n* = 127)	>1 VAP (*n* = 109)	*p*
Age, years ± SD	65 ± 12	63 ± 12	66 ± 12	66 ± 10	0.445
Male, *n* (%)	287 (72.1)	112 (69.1)	93 (73.2)	82 (75.2)	0.517
SAPS II, mean ± SD	40 (31–51)	40 (31–45)	40 (33–48)	42(34–51)	0.369
SOFA, mean ± SD	5 (3–8)	5 (3–7)	5 (3–8)	6 (4–8)	0.479
COMORBIDITIES, *n* (%)					
Chronic Heart Failure	71 (17.8)	28 (17.3)	27 (21.3)	16 (14.7)	0.408
Chronic respiratory failure	48 (12.1)	20 (12.3)	16 (12.6)	12 (11)	0.923
Chronic kidney failure	29 (7.3)	9 (5.6)	12 (9.4)	8 (7.3)	0.450
Hypertension	193 (48.5)	80 (49.4)	58 (45.7)	55 (50.5)	0.731
Diabetes mellitus	140 (35.2)	51 (31.5)	45 (35.4)	44 (40.4)	0.323
Smoker	93 (23.4)	45 (27.8)	22 (17.3)	26 (23.9)	0.114
Obesity	161 (40.5)	72 (44.4)	43 (33.9)	46 (42.2)	0.174
History of neoplasm	42 (10.6)	15 (9.3)	15 (11.8)	12 (11)	0.766
Immunosuppression	39 (9.8)	15 (9.3)	11 (8.7)	13 (11.9)	0.671
Admission periods, *n* (%)					
First wave	67 (16.8)	27 (16.7)	23 (18.1)	17 (15.6)	0.874
Second wave	144 (36.1)	53 (32.7)	48 (37.8)	43 (39.4)	0.474
Third wave	187 (47.0)	82 (50.6)	56 (44.0)	49 (45.0)	0.480
Time from hospital to ICU admission, days, median (IQR)	3 (0–3)	3.2 (0–4)	1.7 (0–2)	2.9 (0–3)	0.68
IMV, *n* (%)	76 (19.1)	29 (17.9)	27 (21.3)	20 (18.3)	0.93
ECMO, *n* (%)	78 (19.6)	22 (13.6)	18 (14.2)	38 (34.9)	**<0.001**
Antiviral agent ^a^, *n* (%)	134 (33.7)	43 (26.5)	46 (36.2)	45 (41.3)	**0.032**
Antibiotic treatment, *n* (%)	264 (66.3)	114 (70.4)	80 (63)	70 (64.2)	0.21
Documented co-infection, *n* (%)	44 (11.1)	12 (7.4)	13 (10.2)	19 (17.4)	**0.035**

Data are presented as median and interquartile range or absolute value and percentage. *p* values in bold were considered statistically significant. ^a^ remdesivir, lopinavir or ritonavir.

**Table 2 jcm-11-07097-t002:** Immunomodulator/immunosuppressive (IS) therapies received during the ICU stay.

	TOTAL (*n* = 398)	0 VAP (*n* = 162)	1 VAP (*n* = 127)	>1 VAP (*n* = 109)
IS therapy, *n* (%)	338 (84.9)	129 (79.6) ^a^	**108 (85)**	**101 (92.7)** ^b^
*Dexamethasone, n (%)*	324 (81.4)	129 (79.6)	103 (81.1)	92 (84.4)
*Methylprednisolone, n (%)*	104 (26.1)	25 (15.4) ^a^	**26 (20.5)** ^a^	**53 (48.6)** ^b,c^
*IL-1 receptor antagonist, n (%)*	15 (3.8)	4 (2.5)	4 (3.1)	7 (6.4)
*JAK receptor antagonist, n (%)*	19 (4.8)	4 (2.5)	8 (6.3)	7 (6.4)
*IL-6 receptor antagonist, n (%)* ^c^	75 (18.8)	**12 (7.4)** ^a,c^	**31 (24.4)** ^b^	32 (29.4) ^b^
*Combination of 2 IS, n (%)* ^d^	193 (48.5)	**51 (31.5)** ^a,c^	**63 (49.6)** ^a,b^	**79 (72.5)** ^b,c^

Values in bold were considered statistically significant. Data are presented as absolute value and percentage. ICU: intensive care unit; IL-1: interleukine 1; IL-6: interleukine 6; IS: immunosuppressive; JAK: janus kinase; VAP: ventilator-associated pneumonia. ^a^
*p* < 0.01 vs. >1 VAP. ^b^
*p* < 0.01 vs. 0 VAP. ^c^
*p* < 0.01 vs. 1 VAP.

**Table 3 jcm-11-07097-t003:** Micro-organisms responsible for VAP (1st episode and recurrences).

	1st VAP (*n* = 338)	2nd VAP (*n* = 165)	3rd VAP (*n* = 69)
Gram-negative bacilli, *n* (%)	189 (55.9)	101 (61.2)	48 (69.6)
*Enterobacteriaceae*	139	60	22
Non-fermenting GNB	50	41	26
Gram-positive cocci, *n* (%)	87 (25.7)	31 (18.8)	8 (11.6)
*MSSA*	50	21	5
*MRSA*	6	3	1
*Enterococcus* spp.	14	6	2
*Streptococcus* spp.	17	1	0
Polymicrobial, *n* (%)	62 (18.3)	33 (20.0)	13 (18.9)
Antibiotic-multiresistant bacteria, *n*	11 (3.2)	14 (8.5)	14 (20.0)
ESBLE-producing Enterobacteriaceae	8	11	10
Carbapenem-resistant *enterobacteriaceae*	1	2	1
Multi-drug resistant *Pseudomonas*	2	1	3

*n* refers to the number of VAP episodes. Data are presented as absolute value and percentage of micro-organisms. ESBLE: extended spectrum beta-lactamase; GNB: gram negative bacilli; MRSA methicillin-resistant Staphylococcus aureus, MSSA methicillin-sensitive Staphylococcus aureus; VAP: ventilator-associated pneumonia

**Table 4 jcm-11-07097-t004:** Clinical outcomes according to the occurrence of VAP and recurrence.

	TOTAL (*n* = 398)	0 VAP (*n* = 162)	1 VAP (*n* = 127)	>1 VAP (*n* = 109)	*p*
OUTCOMES, days, median (IQR)					
Duration of mechanical ventilation	17 (8–36)	10 (5–18)	16 (8–30)	41 (25–56)	**<0.001**
VFD at D28	9 (0–19)	17 (8–22)	11 (0–19)	0 (0–1)	**<0.001**
VFD at D60	41 (21–51)	48 (40–54)	42.5 (28–51)	17 (0–33)	**<0.001**
ICU length of stay	23 (12–42)	14 (9–25)	22 (12–36)	46 (29–66)	**<0.001**
Hospital length of stay	29 (18–49)	22 (14–36)	29 (17–44)	53 (32–75)	**<0.001**
MORTALITY OUTCOMES, *n* (%)					
ICU mortality	111 (27.9)	32 (19.8)	43 (33.9)	36 (33.0)	**0.011**
D28 mortality	69 (17.3)	30 (18.5)	30 (23.6)	9 (8.3)	**0.006**
D90 mortality	114 (28.6)	34 (21.0)	46 (36.2)	34 (31.2)	**0.021**

*p* values in bold were considered statistically significant. Data are presented as median and interquartile range or absolute value and percentage ICU: intensive care unit; IQR: interquartile range; VAP: ventilator-associated pneumonia; VFD: ventilator free days.

**Table 5 jcm-11-07097-t005:** Factors associated with VAP recurrence in univariate and multivariate analysis.

	Univariate Analysis			Multivariate Analysis		
	1 VAP (*n* = 127)	Recurrence (*n* = 109)	*p*	Odds Ratio	95% CI	*p*
Variable						
Age, y	63 ± 12	64 ± 10	0.54			
SAPS 2	40 (33–48)	42 (34–51)	0.13	1	0.98–1.03	0.62
SOFA ^a^	5 (3–8)	6 (4–8)	0.20			
Obesity, *n* (%)	43 (34)	46 (42)	0.22			
IS treatment (at least one), *n* (%)	106 (83)	101 (93)	0.06	0.5	0.18–1.39	0.19
Steroidal IS, *n* (%)	42 (33)	74 (68)	<0.001	0.75	0.37–1.52	0.43
Non steroidal IS, *n* (%)	39 (31)	39 (36)	0.46			
Association of 2 IS, *n* (%)	62 (49)	79 (73)	<0.001	0.66	0.34–1.27	0.21
Bacterial co-infection at ICU admission, *n* (%)	12 (9)	19 (17)	0.08	0.64	0.26–1.56	0.32
Antibiotic target attainment ^b^	20 (69)	30 (75)	0.07	0.94	0.43–2.09	0.89
Duration of MV	16 (8–30)	41 (25–56)	<0.001	1.06	1.04–1.08	**<0.001**
First VAP documentation						
Gram positive Cocci	51	42	0.72			
*Enterobacteriaceae*	60	61	0.22			
*Non-fermenting negative Gram Bacilli*	24	23	0.72			

*p* values in bold were considered statistically significant. Quantitative variable are presented as mean ± standard deviation or median and interquartile range. ^a^ On the day of ICU admission. ^b^ *n* = 29 for 1 VAP group and *n* = 40 for recurrence group. CI: confidence interval; ICU: intensive care unit; IS: immunosuppressive/immunomodulatory; MV: mechanical ventilation; SAPS2: Simplified Acute Physiology Score 2.

## Data Availability

The datasets used and/or analyzed during the current study are available from the corresponding author on reasonable request.

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
