# Peer review of "Incidence, Outcomes and Risk Factors of Recurrent Ventilator Associated Pneumonia in COVID-19 Patients: A Retrospective Multicenter Study"

_jcm, 2022, doi:10.3390/jcm11237097_

Round 1
Reviewer 1 Report
From a methodological point of view manuscript is somewhat confusing, in method section the authors declare things that they do not use and the results presented are incomplete and poorly curated.
The authors write that the primary outcome was the incidence of VAP relapses but do not present an incidence analysis of the 236 subjects who developed VAP. The sample has a 90-day mortality of 28.6% and I do not think it is the correct way to assess the incidence of VAP without considering the competitive risk of death.
In method section authors mention mstate R package that contains functions for prediction with Aalen-Johansen or simulation in competing risks, have you used any functions such as?
What are the covariates used in the Cox model to confirmed impact of variables on timing of VAP incidence? The authors are strongly encouraged to include details of how the proportionality assumption in Cox model was assessed and confirm there were no gross violations.
Please provide details on follow-up data collection for survival. How many patients were lost to follow-up? What was the date of the last census? The risk set should be added under each curve. Furthermore at Figure 2a, one of the two curves has no asymptote. Please, can you explain why it is truncated?
The results in the tables are poorly curated. It is not a good practice to report only the p-values of significant statistical tests. For transparency and completeness, all the tables should be completed with the calculated p-values even if > 0.05, also the legend of Table 1 is not exhaustive,…
At table 2, why were the statistical summaries not tested?
Table 3 should be further explained: the number 338 refer to patient or VAP episode?
Please, present score (SOFA, SAPSII,…) also as median and first and third quartile.
Author Response
Journal name: Journal of Clinical Medicine
Manuscript ID: jcm-1990625
Type of manuscript: ArticleTitle: Incidence, outcomes and risk factors of recurrent ventilator associated pneumonia in COVID-19 patients: a retrospective multicenter studyAuthors: Ines Gragueb-Chatti *, Hervé Hyvernat, Marc Leone, Geoffray Agard, Noémie Peres, Christophe Guervilly, Mohamed Boucekine, Dani Hamidi, Laurent Papazian, Jean Dellamonica, Alexandre Lopez, Sami Hraiech
Dear Editor and Reviewers,
Thank you for offering us the possibility to submit a revised version of our manuscript entitled “Incidence, outcomes and risk factors of recurrent ventilator associated pneumonia in COVID-19 patients: a retrospective multicenter study”.We would like the Editor and Reviewers for their relevant comments that helped us to enhance the quality of our manuscript.
Please find below a point-by-point response to Editor and Reviewers comments. All changes are highlighted in yellow in our revised version. Please see the attachment.
Best regards.
Reviewer 1:
Reviewer’s comment
From a methodological point of view manuscript is somewhat confusing, in method section the authors declare things that they do not use and the results presented are incomplete and poorly curated.
Authors’response:
We agree with the reviewer that the methods section and particularly the statistical analyses described were somewhat confusing. We clarified this section and removed some elements finally not used but added the cumulative incidence that was suggested by reviewer (see below). The results section has been modified consequently.
Reviewer’s comment
The authors write that the primary outcome was the incidence of VAP relapses but do not present an incidence analysis of the 236 subjects who developed VAP.
The sample has a 90-day mortality of 28.6% and I do not think it is the correct way to assess the incidence of VAP without considering the competitive risk of death.
Authors’response:
We thank the reviewer for this suggestion. To answer this question, we calculated the incidence of VAP recurrences considering death and extubation as competitive risk. Fine and Gray model was used to estimate the cumulative incidence. The cumulative incidence of VAP recurrence was 29,6 (IC [25 – 34 ])%
These elements have been added into the methods and results sections. A figure representing these data has been added (figure 2).
Reviewer’s comment
In method section authors mention mstate R package that contains functions for prediction with Aalen-Johansen or simulation in competing risks, have you used any functions such as?
Authors’response:
We thank the reviewer for this remark.
Analysis derived from the mstate were first performed but finally not included because data concerning VAP recurrences after the 2nd episode were incomplete. We removed this sentence from the manuscript.
However as suggested, we performed a complementary analysis to consider death and extubation as competing risk of VAP recurrence. Fine-Gray model was used to estimate the cumulative incidence. Analysis was performed using the cuminc() function from cmprsk r package.
Reviewer’s comment
What are the covariates used in the Cox model to confirmed impact of variables on timing of VAP incidence? The authors are strongly encouraged to include details of how the proportionality assumption in Cox model was assessed and confirm there were no gross violations.
Authors’response:
We thank the reviewer for this suggestion.
We used univariate Cox model testing :
- all immunosuppressive (IS) therapies
- only steroidal IS or
- only non-steroidal IS
on the delay of VAP relapse.
The assumption of proportional hazards of Cox model was assessed using the produced product of the time (delay of VAP relapse) by the covariable.
Concerning the produced product of the time (delay of VAP relapse) by the covariable, p value was respectively 0.276 for all IS, 0.923 for steroidal IS and 0.220 for non-steroidal IS indicating for all models no gross violation of proportional hazard of Cox model.
These results have been detailed in the manuscript line 276.
Reviewer’s comment
Please provide details on follow-up data collection for survival. How many patients were lost to follow-up?
Authors’response:
Follow-up data collection was conducted by analyzing the medical files or calling patients or their relatives.
At the time of final analysis, 19 patients were lost to follow-up.
This information has been added in the results section in line 240.
Reviewer’s comment
What was the date of the last census?
Authors’response:
The last census date was August 2021, 90 days after the last inclusion.
Reviewer’s comment
The risk set should be added under each curve. Furthermore at Figure 2a, one of the two curves has no asymptote. Please, can you explain why it is truncated?
Authors’response:
We thank the reviewer for his comment. The risk set has been added under each curve.
In figure 2a (which is now referred as Figure 3a), one of the two curves has indeed no asymptote because all the patients without steroidal IS treatment had deloped a VAP relapse after the 30th day of mechanical ventilation.
Reviewer’s comment
The results in the tables are poorly curated. It is not a good practice to report only the p-values of significant statistical tests. For transparency and completeness, all the tables should be completed with the calculated p-values even if > 0.05, also the legend of Table 1 is not exhaustive,…
Authors’response:
Variable manners of presenting the results in tables can be encountered and some reviewers prefer not to present the p value when not significant, to avoid burdening. However, we followed the reviewer’s remark and added all the p-values for transparency in table 1. The legend has also been completed.
Reviewer’s comment
At table 2, why were the statistical summaries not tested?
Authors’response:
Statistical were tested however, for more clarity, we chose to show the comparison tests between each group (indicated by letters in the table for significant p values). We thought this presentation might be easier to understand for the reader.
Reviewer’s comment
Table 3 should be further explained: the number 338 refer to patient or VAP episode?
Authors’response:
Indeed, the number 338 refers to VAP episodes. This has been clarified into the table legend.
Reviewer’s comment
Please, present score (SOFA, SAPSII,…) also as median and first and third quartile.
Authors’response:
Scores are now presented as median and quartiles in tables 1 and 5, as requested.
We thank the reviewer for his very relevant comments.

Reviewer 2 Report
Chatti et. al report on a retrospective cohort of patients with severe COVID19 requiring mechanical ventilation and describe the incidence and outcomes of ventilator associated pneumonia. The methodology is sound with well-defined clinical endpoints and assessment of covariates. The results incrementally add to the body of literature around VAP in COVID19.
I disagree with the author’s conclusions about their data. The observed risk factors for VAP (Duration of MV, IS use) are strongly associated with severity of illness. Indeed, persistent mechanical ventilation is a requirement for developing VAP. I do not think that the analysis in it’s current form is suited to address the causal link of IS and VAP (line 283).
I also have several minor statistical comments. Given the number of comparisions made, some attempt should be made to adjust for multiplicity. The survival analysis (Figure 2a and 2b) is not totally described. When were subjects censored? What parameter is estimated using a Cox model? (I see a Kaplan-Meier curve and log rank test but no cox estimator). Are the assumptions met for the estimator used (For K-M: Is the censoring risk the same for both groups? Are there secular trends in the outcome?).
Author Response
Journal name: Journal of Clinical Medicine
Manuscript ID: jcm-1990625
Type of manuscript: ArticleTitle: Incidence, outcomes and risk factors of recurrent ventilator associated pneumonia in COVID-19 patients: a retrospective multicenter studyAuthors: Ines Gragueb-Chatti *, Hervé Hyvernat, Marc Leone, Geoffray Agard, Noémie Peres, Christophe Guervilly, Mohamed Boucekine, Dani Hamidi, Laurent Papazian, Jean Dellamonica, Alexandre Lopez, Sami Hraiech
Dear Editor and Reviewers,
Thank you for offering us the possibility to submit a revised version of our manuscript entitled “Incidence, outcomes and risk factors of recurrent ventilator associated pneumonia in COVID-19 patients: a retrospective multicenter study”.We would like the Editor and Reviewers for their relevant comments that helped us to enhance the quality of our manuscript.
Please find below a point-by-point response to Editor and Reviewers comments. All changes are highlighted in yellow in our revised version. Please see the attachment.
Best regards.
Reviewer 2 :
Reviewer’s comment
Chatti et. al report on a retrospective cohort of patients with severe COVID19 requiring mechanical ventilation and describe the incidence and outcomes of ventilator associated pneumonia. The methodology is sound with well-defined clinical endpoints and assessment of covariates. The results incrementally add to the body of literature around VAP in COVID19.
Authors’response:
We thank the reviewer for these remarks.
Reviewer’s comment
I disagree with the author’s conclusions about their data. The observed risk factors for VAP (Duration of MV, IS use) are strongly associated with severity of illness. Indeed, persistent mechanical ventilation is a requirement for developing VAP. I do not think that the analysis in it’s current form is suited to address the causal link of IS and VAP (line 283).
Authors’response:
In our multivariate analysis including age, SOFA, duration of MV, bacterial co-infections, bacterial documentation, and immunosuppressive treatments, IS was not associated with a higher rate of VAP recurrence. However, we agree with the reviewer’s on the methodological limits highlighted that precludes to firmly conclude about the role of IS treatments. This has been added in the discussion section as a study limitation.
Reviewer’s comment
I also have several minor statistical comments. Given the number of comparisions made, some attempt should be made to adjust for multiplicity.
Authors’response:
We thank the reviewer for this suggestion.
We used univariate Cox model testing : a) all IS therapies, b) only steroidal IS or c) only non-steroidal IS on delay of VAP relapse.
The assumption of proportional hazards of Cox model was assessed using the produced product of the time (delay of VAP relapse) by the covariable.
Concerning the produced product of the time (delay of VAP relapse) by the covariable, p value was respectively 0.276 for all IS, 0.923 for steroidal IS and 0.220 for non-steroidal IS indicating for all models no gross violation of proportional hazard of Cox model.
These results have been detailed in the manuscript line 276.
Reviewerr’s comment
The survival analysis (Figure 2a and 2b) is not totally described. When were subjects censored?
Authors’response:
Subjects were censored at the time of VAP recurrence. This has been added in the figure legend.
Reviewer’s comment
What parameter is estimated using a Cox model? (I see a Kaplan-Meier curve and log rank test but no cox estimator).
Authors’response:
We used univariate Cox model testing a) all IS therapies, b) only steroidal IS or c) only non-steroidal IS on delay of VAP relapse. The hazard ratio of Cox model was respectively a) 1.074 (IC [0.493-2.338]), b) 0.546 (IC [0.359-0.828]), c) 1.980 (IC [1.275-3.074]).
Reviewer’s comment
Are the assumptions met for the estimator used (For K-M: Is the censoring risk the same for both groups? Are there secular trends in the outcome?).
Author’s response: in Kaplan Meyer curves, the censoring risk was the same in both groups.
We acknowledge the reviewer for his helpful comments.

Round 2
Reviewer 2 Report
The authors have appropriately responded to the comments and addressed concerns.